# Determination of Critical Power Using Different Possible Approaches among Endurance Athletes: A Review

**DOI:** 10.3390/ijerph19137589

**Published:** 2022-06-21

**Authors:** Lucie Lipková, Michal Kumstát, Ivan Struhár

**Affiliations:** Department of Health Promotion, Faculty of Sports Studies, Masaryk University, Kamenice 5, 625 00 Brno, Czech Republic; kumstat@fsps.muni.cz (M.K.); struhar@fsps.muni.cz (I.S.)

**Keywords:** critical work, methods, performance analysis, endurance athletes

## Abstract

Critical power represents an important parameter of aerobic function and is the highest average effort that can be sustained for a period of time without fatigue. Critical power is determined mainly in the laboratory. Many different approaches have been applied in testing methods, and it is a difficult task to determine which testing protocol it the most suitable. This review aims to evaluate all possible tests on bicycle ergometers or bicycles used to estimate critical power and to compare them. A literature search was conducted in four databases (PubMed, Scopus, SPORTDiscus, and Web of Science) published from 2012 to 2022 and followed the PRISMA guidelines to process the review. Twenty-one articles met the eligibility criteria: records with trained or experienced endurance athletes (adults > 18), bicycle ergometer, a description of the testing protocol, and comparison of the tests. We found that the most widely used tests were the 3-min all-out tests set in a linear mode and the traditional protocol time to exhaustion. Some other alternatives could have been used but were not as regular. To summarize, the testing methods offered two main approaches in the laboratory (time to exhaustion test andthe 3-min all-out test with different protocols) and approach in the field, which is not yet completely standardized.

## 1. Introduction

Critical power (CP) represents the boundary between the intensity domains of heavy and severe exercise [1]. It is the greatest average effort that can be sustained for a period of time without fatigue. Endurance athletes are exposed to a fast pace, but even a slight increase in pace may lead to a reduction in the tolerable duration of exercise. This is what CP can predict [2]. From the mathematical viewpoint, CP is defined as the power-asymptote of the hyperbolic relationship between power output (PO) and time-to-exhaustion [3]. The power-asymptote stands for CP and the hyperbole stands for the amount of work done above CP (W’); together they predict the tolerable duration of exercise above the CP [4]. Figure 1 shows the power–time (P-t) relationship for high-intensity exercise. The area above the CP asymptote but under the curve represents the amount of work (in joules) over CP that can be performed until exhaustion occurs. Each grey shaded box shows how much work can be performed at specific PO with the CP, in this case 300 W. The important thing is that every athlete has a unique critical power curve, which has to be constructed from their own individual exercise tests.

W’ is constant, but may be utilized at different rates depending on the proximity of the exercise PO to the CP [5]. W’ is measured in kilojoules (kJ) and is a function of the oxygen uptake (VO_2_) slow component; meaning a slow increase in the VO_2_ during constant work performed above the lactate threshold [6]; maximal oxygen uptake (VO_2max_); the depletion of intramuscular substrates, such as muscle creatine phosphate; and the glycogen, and the accumulation of metabolites (hydrogen ions, adenosine diphosphate, phosphate ions), which is associated with impaired muscle contractile function [7]. The CP represents an important parameter of an aerobic function. It can provide an even more meaningful criterion based on the measure of external PO and time than the lactate threshold or the maximal uptake of oxygen (O_2_) [3,8]. Rather than using equivalent exercise intensities relative to these metabolic parameters (lactate threshold, VO_2max_), setting exercise intensity relative to CP combines systemic and intramuscular response, and the W’ is expended above the CP, VO_2max_ is attained, and intolerance is manifested. The CP occurs approximately at 70–80% VO_2max_ and trained individuals can reach 80–90% VO_2max_ [4]. 

This has a significant potential for sports performance but is often taken as a purely mathematical construct that does not have a physiological importance [3]. The CP is primarily the rate of oxidative metabolism rather than the mechanical PO (by which it is typically measured). This can properly be termed “critical VO_2_”. In cycling, the PO corresponds to this critical VO_2_ that can be altered with the chosen pedal rate. The actual CS (critical speed equivalent to CP) is also equivalent to the critical VO_2_ but depends on the movement economy. This is because critical VO_2_ is expressed functionally in units of power or speed that are effective in the prediction of exercise tolerance or performance [4].

Understanding CP can help with racing strategies. This concept is best explained through considering the performance of two runners with an identical gas exchange threshold (GET), which is used as an index of anaerobic threshold [9] and VO_2max_ values. Using the metric of running economy (i.e., the percentage of GET relative to VO_2max_), the runners may be considered identical. However, one runner has a higher CS and lower D’ (distance equivalent to W’) compared to the other. The runner with higher CS should adopt a front-running strategy rather than holding back. This strategy involves running faster in split-times rather than trying to store energy for a big finish. This means that running or cycling below the CS/CP will result in a slower time over a middle distance. On the other hand, the runner with a higher D’ should try to keep the pack of runners closer to his pace and then rely on a large portion of D’ for a big finish [10]. Once you know your CP and W’, you can predict the maximal power you could sustain over various periods of time, and this can be useful for creating pace strategies. 

The CP concept is mainly generalized to endurance sports, such as running, rowing, swimming, and cycling, and continuous and intermittent isometric exercise. This model has been modified for intermittent exercise and has a potential application in interval training and team sports, such as football, rugby, hockey, etc. It cannot be applied to sporting activities that involve a single or only a few muscle contractions (e.g., field athletics, archery), sports where the work-to-rest ratios are such that the limits set by the W’ parameter are unlikely to be challenged (e.g., American football, baseball, cricket) or events where the PO does not exceed the CP (e.g., golf, ultra-endurance events) [3]. The CP is typically determined in the laboratory, though it is also possible to use field conditions in the form of cycling in the velodrome or on the road for a certain time [5]. Recent research found that testing maximal power indoors and outdoors cannot be used interchangeably. It is hard to translate results from indoor testing to outdoor among elite cyclists due to individual variation [11]. Conditions such as cadence, body position, level ground, or uphill conditions have been shown to have an impact on the estimation of the critical power. However, a good agreement exists between some laboratory tests and real cycling performance. Testing protocols should mirror as closely as possible the competition settings to provide environmental validity. As an example, climbing specialists should perform prediction trials on a road bike in uphill conditions, while time trials specialists should conduct testing on a time trial bike on level ground [12].

The advantage of CP testing is that it gives you two physiological markers from one test (CP and W’). Knowing both the CP and W’ can be useful in determining how to balance these different parts of the physiology, similarly to how the lactate threshold is a balance between the aerobic and anaerobic systems [5]. Due to the different approaches to testing methods, it is a difficult task to determine which testing protocol is the most suitable. There is a large diversity of methods and many criteria can be chosen to establish the testing protocol. This review searched the more recent scientific literature from the years 2012–2022 to provide a comprehensive overview of testing protocols. The aim of this paper is to evaluate all possible tests on bicycle ergometers and their alternatives in order to determine CP among endurance athletes and compare them. This could simplify decisions relating to test selection and provide a compact overview of all possible criteria that affect testing.

## 2. Materials and Methods

### Search Strategy

A literature search was conducted in four databases: PubMed, Scopus, SPORTDiscus, and Web of Science. The keywords put into the search were: (“critical power”) AND (“test” OR “tests” OR “method” OR “methods”) AND (“bicycle ergometer” OR “cycle” OR “cycling”). This review followed the PRISMA flow diagram (Figure 2). Several records (*n* = 4) were added from the literature search found in the relevant articles. 

Exclusion criteria were defined for the search strategy to exclude unwanted studies: duplicates;language (if not English);wrong study design: review articles, meeting abstracts, letters, corrections, and editorial materials;participants (if not human);no words (if they did not include keywords);wrong words (critical speed, critical velocity, critical forces, critical torque);year (if not 2012–2022).

The remaining records were assessed for eligibility and were not included if they did not comply with the inclusion criteria. The criteria that had to be met were: trained or experienced endurance athletes, adults (>18), detailed description of the test (bicycle ergometer, number of visits laboratory/field, resistance, cadence, calculation of CP and W’), comparison of the tests, and number of participants (*n* > 6). Studies had to fulfil these criteria to be included in this review. 

A total of 1114 studies were identified from the databases (Figure 2). Before screening, 622 records of duplicates were eliminated A total of 170 full-text records were screened, though only 21 studies were included in the review. The 149 pieces of research that did not match the eligibility criteria based on full-text screening were excluded because of the aforementioned inclusion criteria. 

## 3. Results

Twenty-one studies remained for more detailed analysis. All of these are concerned with the same major subject—CP. The studies contain different methods, conditions (laboratory, field), resistance/cadence, cycle ergometer, and choice of mathematical model.

### 3.1. Characteristics of the Participants 

The characteristics of the participants are summarized in Table 1. In total, 242 participants were involved in the records (199 males, 43 females, mean ± SD: age = 29.1 ± 5.9 years). Ten of the studies included men and women [13,14,15,16,17,18,19,20,21,22], while the rest included only men [8,23,24,25,26,27,28,29,30,31,32]. The inclusion criteria for participants were defined as trained and experienced athletes. Four studies described moderate and recreationally trained subjects, which is somewhat in conflict with these criteria [13,14,17,25], but all participants were capable of completing the test and all studies contain important information. The other studies described subjects as elite/competitive, club-level, trained, or a combination [8,15,19,20,22,23,24,26,28,29,30,31,32]. Three incorporated various fitness levels that included competitive and recreationally trained individuals [16,18,21]. All participants were focused on endurance sports, mainly cycling. 

### 3.2. Classification of the Testing Protocols 

The studies that are included in this review contain laboratory and field testing. Laboratory testing is subdivided in two main types (Figure 3). The first method is the time-to-exhaustion test (TTE), which is the original method. Another option based on this protocol is the time trial (TT), which attempts to represent real-world conditions. A more recent protocol is the 3-min all-out test (3MT) set in a linear mode, which means against a fixed resistance. The other possibility is 3MT in an isokinetic mode (3MT_iso_) with fixed cadence or a protocol with resistance set by % body mass (% BM) (3MT_%BM_). Field testing is performed over a fixed time or involves collecting data from uncontrolled conditions.

#### 3.2.1. Traditional Laboratory Testing Protocol 

The traditional testing protocol requires a series of 3–5 (TTE) trials at severe constant work rates on separate days. These tests should last between 1 and 20 min and are usually separated by 24 or 48 h. Before the testing protocol, a preliminary maximal ramp or graded incremental test is completed for the determination of maximal VO_2_ and peak power output (PO_peak_). In this review, all testing protocols are performed on an electronically braked cycle ergometer. The work rates are estimated from previous tests and subjects should maintain their preferred or given cadence for as long as possible. A test is terminated when the cadence falls to more than 5 or 10 revolutions per minute (rpm) below the preferred cadence for more than 5 or 10 s. It is appropriate to provide strong verbal encouragement throughout the test. Subjects are not informed of the work rates or their performance on any of the tests [14,21,33]. 

There is an interest in determining the power–duration relationship of athletes and to here this is conducted more practically with a scientifically valid and time-efficient assessment method. This method requires only a power meter and stopwatch, and, with the addition of a stationary trainer, it can be performed anywhere, at any time, with laboratory-style reproducible conditions. Using a series of self-paced, fixed-duration TT, in which the objective is to average the highest PO possible for a given duration, CP and W’ can be obtained over several days or in just a single visit [22,24]. During a TT attempt, the PO does not necessarily approach maximal values but may also fluctuate throughout, while the PO for TTE attempts is driven up to a pre-determined fixed intensity in a square-wave. Different cadences in TTE and TT attempts may produce different CP values. However, both TT and TTE attempt intensities are located in the severe domain. W′ depletes towards zero in all trial types independently of related power profiles [24]. 

The estimation of CP can vary and is influenced by the testing protocol. Factors that can change the calculation of the CP are the mode and duration of the test, cadence, resistance, the ergometer, and the mathematical model [21]. TTE and TT trials can determine the CP using different mathematical models. The first model, the linear total work (Linear-TW) is based on the linear regression of TW vs. *t*:TW = AWC + CP × *t*(1)
where *t* is the time to exhaustion, TW is the total work (P×t), AWC is the anaerobic work capacity, and CP is the critical power. The second model is a linear model (Linear-P or 1/time) derived for *P*, using two equations for the total work, TW = *P* × *t*, and TW = AWC + CP × *t* to obtain the equation:*P* = AWC × (1/*t*) + CP,(2)
where 1/t is the inverse of T and *P* is the PO. 

The third mathematical model is based on the non-linear hyperbolic relationship between power *P* and *T* to obtain a 2-hyp model: *t* = AWC/(*P* − CP).(3)

The fourth model is a nonlinear model that attempts to estimate CP more accurately by including a parameter for the maximal instantaneous power (*P*_max_). The estimate of *P*_max_ and the derivation of the 3-hyp model are made possible by the addition of a third parameter *k*, which allows a non-zero time asymptote at *t* = *k* to obtain the final equation:*t* = (AWC/(*P* − CP)) − (AWC/(*P*_max_ − CP)).(4)

The fifth and final regression model is the exponential (EXP): (5)$$P=\mathrm{CP}+(P_{\max}-\mathrm{CP})\;\times \;\exp\;(-t/\tau ),$$
where τ is an undefined time constant. This model does not estimate W’ [14,21,34].

#### 3.2.2. New Testing Protocols in Laboratory (3-Min All-Out Tests) 

The 3MT consists of cycling maximally against a fixed resistance (linear mode), during which participants should not pace themselves. This mode is set using the following equation: linear factor (resistance) = power output/preferred cadence^2^,(6)
where PO is determined as the midway point between the GET and peak oxygen uptake (VO_2peak_). These variables (GET, VO_2peak_ and cadence) are estimated from a preliminary incremental graded or ramp test to exhaustion. The test was originally carried out using a Lode Excalibur sport cycle ergometer [28]. Testing provides a metabolic signal to the mitochondria to elevate respiration so that VO_2_ increases rapidly to VO_2max_ and remains there as power falls to CP with W’ being completely utilized. This provides an additional cost of O_2_ corresponding to a major loss of efficiency, which is similar to the development of the slow component of oxygen uptake during a constant work rate throughout severe intensity exercise leading to exhaustion [33,35]. After the complete depletion of W’, this results in PO being maintained from only aerobic metabolism. This end power is equal to the CP, and work above end power (WEP) equals W’. The next procedure for establishing the load for the 3MT is set by using a % BM based on the subject’s fitness level [13,16,18]. 

An alternative option is testing in the isokinetic mode, which is performed against a constant cadence. In this protocol, participants can cycle at their preferred cadence or a cadence chosen by researchers. In this mode, subjects are not allowed to cycle faster than the selected cadence but are encouraged to cycle at maximal effort throughout the test. The resistance is adapted to any change in pedal force to maintain chosen cadence [20,28]. The CP of all these approaches is estimated from the mean power of the last 30 s of the all-out trial, where W’ can be estimated using the following equation: W’ = 150 × (*P*_150_ − CP), (7)
where *P*_150_ is the mean power for 150 s of all-out exercise [16].

#### 3.2.3. Field Tests 

The determination of the CP in cycling is traditionally based in the laboratory [19]. The increasing utilization of mobile power meters in cyclists during training and racing has focused greater attention on estimation of the power–duration relationship based on field-derived data. Since the development of a single-visit protocol separated by a recovery time of 30 to 60 min, it has been transferred to the field where the CP and W’ derived from maximal-effort trials have been validated against parameter estimates from traditional multi-visit TTE trails under laboratory conditions. Linear regression analyses between the CP and performance (e.g., time, power, or speed) can be used to access predictive validity and thereby neglect W′ as an important contributor to an athlete’s maximal work capacity. The field test consists mostly of three maximal efforts over a fixed duration or fixed distance during which subjects should produce the highest possible power. PO is measured with a power meter and participants can choose a preferred cadence and gear ratio. The duration of these trials and mean power is used to estimate the CP and W’ using the mathematical models mentioned above (Section 3.2.1), [26,27]. 

## 4. Discussion

This review demonstrates all known possible methods of determining the critical power (CP) on bicycle ergometer or on the bike in the field and provides a comprehensive overview. In general, the methods used can be divided into traditional tests and 3MT. In the case of traditional testing, the constant power TTE is the most widely used. The next possibility is to use a self-paced TT. Both tests are recognized for determining cycling performance and are used to estimate critical power from a series of 3–5 tests until exhaustion [13,14,19,20,21,28,29,30]. 

The original test requires a large number of trials performed on separate days, but there is also the option of using the traditional protocol with testing within one day with different recovery times ranging from 30 min up to 3 h [23,24,25,27,30]. This may help make the test more readily available and usable in practice. Despite the simplification of a single laboratory visit, this still involves a larger number of tests, which is very demanding for the participants themselves. Due to this, the 3MT protocol was introduced as a single test that requires only preliminary testing to determine PO, GET, and VO_2max/peak_. The standard 3MT test is set in a linear mode, which means that it is performed at a fixed resistance [8,13,14,16,17,28,29]. An alternative possibility is 3MT_iso_ with a fixed cadence [18,19,29,36]. Despite the existence of two tests (preliminary and 3MT), another option has also been developed-3MT_%BM_ [13,16,18]. This option requires only one test with one visit to the laboratory. Another possibility is to determine the CP and W’ from tests performed in the field [19,23,26,27,31,32]. 

Different mathematical models can be used for the determination of the CP from traditional protocols (TTE, TT). All of these models (linear total work, linear 1/time, 2-parameter hyperbolic, 3-parameter hyperbolic, and exponential) were compared by Bergstrom et al., Clark and Macdermid, and Maturana et al. in their studies [14,21,32].


### 4.1. The Traditional Protocol and Its Possibilities 

As mentioned, the traditional protocol TTE typically requires 3–5 tests that are standardly performed on separate days. This testing needs a preliminary incremental test to exhaustion to determine VO_2peak_/VO_2max_ or maximal aerobic power (MAP) and the GET [13,19,20]. This test is necessary to set the resistance and cadence for the TTE protocol. The TTE is based on pedaling at various resistances until exhaustion. Due to the time constraint, Karsten et al. [25] researched different recovery times between each test (24-h recovery, 3-h recovery, and 30-min recovery). In this study, resistance was set at 80, 100, and 105% of power at VO_2max_ (pVO_2max_), with the preferred cadence of the subject being achieved on a bike with a PowerTap (CycleOps, Madison, WI, USA). The results suggested that the CP can be determined using a recovery period of 3 h or 30 min due to high agreement and low prediction error. This could significantly speed up the entire testing and make it more readily available. 

#### 4.1.1. Comparing Different Mathematical Models 

Five mathematical models are described for calculating the CP and W’. Two studies have compared all these models and one study compared four models (Linear-TW, Linear-P = 1/time, 2-hyp, and 3-hyp) and determined the most suitable [14,21,32]. The study by Bergstrom et al. [14] used a protocol of four TTE trials, as compared with the five TTE trials used by Maturana et al. [21]. Bergstrom et al. [14] set the resistance to 70%, 80% ∆ (i.e., GET + 70% ∆ and 80% ∆, where ∆ was the magnitude of the interval between GET and VO_2peak_) and 100%, 105% VO2_peak_ on an electronically braked Lode cycle ergometer (Corival, Groningen, the Netherlands) with a cadence of 70 rpm. In the study by Maturana et al. [21], tests were performed at 80%, 95%, and 110% of PO_peak_, with the power for the other two tests being determined so as to generate an even distribution of TTE between the five trials. Participants cycled at a preferred cadence (70–105) on an electromagnetically braked cycle ergometer (Velotron Dynafit Pro, Racer Mate, Seattle, WA, USA). The tudy by Clark and Macdermind used TT trials (1, 4 and 10 min) conducted on their own bicycle fitted to a smart trainer. Participants were required to ride as fast as possible in all trials (self-selected pacing strategy) [32].

Table 2 summarizes the results of these three pieces of research. The results for the CP are compared with each mathematical model and an assessment is made as to whether the result is greater or smaller than the values from other mathematical models. All studies obtained similar results among CP. The highest value was determined from EXP [14,21] or 1/time [32] and, in contrast, the lowest from the 3-hyp model. One difference was noted between these studies. Bergstrom et al. [14] found values from 3-hyp and 2-hyp that were almost identical, while Maturana et al. [21] found a greater difference between them with a higher value from 3-hyp. The estimation of W’ varied more between each study [14,21,32].

Linear, EXP, and CP_3min_ overestimated the CP, while the W’ was underestimated in these models. CP and W’ determined from the hyp-3 model seemed to estimate the asymptote of power–duration accurately [14]. The results confirmed Maturana et al. [21] claimed that accurate estimations of CP may be performed with fewer tests and simpler analyses with the 2-hyp, linear-TW, and 1/time models. However, it is essential that these trials last from approximately 7 to 20 min. If the trials last less than 10 min, this may lead to the overestimation of CP, especially with the use of the linear and 1/time models. The results from Clark and Macdermind [32] supported the linear compared to nonlinear models. This contradicts findings by Maturana et al. [21] and Bergstrom et al. [14].

#### 4.1.2. Another Testing Possibility (Time-Trial) 

The next possibility is to use the traditional protocol with a fixed time, resulting in more practical time efficiency and a scientifically valid method. PO fluctuates throughout testing over a specific duration during this TT protocol. Karsten et al. [24] applied 12, 7, and 3 min, in that order. The study mentioned above used trials 1, 4, and 10 min [32]. The resistance increased as a function of cadence and pedal force at the start of each TT. Karsten et al. [24] compared this method with TTE consisting of three trials with a work equivalent to 85%, 100%, and 105% MAP, from lowest to highest in order. Both trials contained a recovery time of 60 min between each test. A personal racing or TT bike mounted on a Cyclus2 ergometer (RBM Electronics, Leipzig, Germany) was used and a 1/time model was chosen. The TT protocol was no different from the TTE, and a high level of agreement and a low prediction error was found. A finding by Coakley and Passfied [30] was that average PO for TTE was higher when compared to the TT at 80% but not at 100% and 105% MAP. This test was conducted on a cycle ergometer (Computrainer Pro, Racer Mate Inc, Seattle, WA, USA) on separate days with at least 48 h between each test. The TTE protocol was same as by Karsten et al. [25] with 30 min recovery between each trial. The final visit included three TT of the same duration, as previously recorded for the TTE trials at each PO using linear-TW and 1/time. Participants were free to change their cadence and ergometer resistance to complete as much work as possible. The result for the lower TT performance at 80% MAP may be related to competitive cyclists’ pacing strategy by starting too fast [30].

Simpson and Kordi [22] attempted to verify a simplification of this protocol in their research. They examined 2 and 3 TT methods on their own road racing bicycle fitted with a power meter (PowerTap G3 wheel; CycleOps, Madison, WI, USA) attached to a custom-made, air-resisted stationary trainer. The 2-trial contained 3-min and 12-min tests with a 30-min time-to-recovery and 10 min for re-warm-up. A 5-min TT was performed separately to complete the 3-trial. The purpose of the test was similar to that given by Karsten et al. [24], i.e., to finish the trial with all possible effort with nothing left to give with a freely changing cadence. The CP and W’ values were estimated from linear and 1/time models. This study concluded that CP and W’ can be determined from just two 3-min and 12-min trials, further simplifying the method. 

### 4.2. 3-Min All-Out Tests

This method was introduced to simplify the test as compared to a traditional protocol that requires multiple tests. This test assumes that the anaerobic capacity is exhausted throughout the test and that the final performance is sustainable only by the aerobic system. The end power (EP) represents the average power during the last 30 s and corresponds to the CP. Work above the EP (WEP), that is the average power over 150 s, corresponds to W’ [13]. The original 3MT is traditionally performed on a Lode Excalibur Sport cycle ergometer with fixed resistance set in the linear mode with a preliminary test to set the resistance [13,16,28,29]. The aim is to maintain a cadence as high as possible throughout the test against a fixed resistance. It can be conducted on other types of ergometers and in a different mode or with different resistance. 

#### 4.2.1. The Original Test (3MT_lin_)

The accuracy of the test was evaluated in research conducted by Wright et al. [29] in which 3MT was compared with TTE with three separated trials at 80%, 100%, and 105% MAP, calculated using a 1/time model. The 3MT trial was set in a linear mode (3MT_lin_) on a Lode Excalibur Sport ergometer. The cadence was manipulated, and the results suggested that the 3MT test is affected by the cadence and a selected cadence 10 rpm above the preferred cadence provided the closest estimate of CP. W’, in contrast, was not affected by the cadence but was significantly underestimated at all cadences (+5, −5, +10). Bergstrom et al. [14] performed a 3MT on a Lode Corival electronically braked cycle ergometer in a linear mode with a fixed resistance. The results showed an overestimation of CP and an underestimation of W’ estimated by the 3MT, linear, and EXP models. The TTE contained four trials with constant power equal to 70% and 80% ∆ and 100% and 105% VO_2peak_ with a cadence of 70 rpm. Similar trends were reported by Bergstrom et al. [13] in their work comparing 3MT_lin_ and TTE. The results of the 3MT_lin_ test were greater than those of the TTE, but there were no significant mean differences between the W’ values (AWC). This study determined the CP on a Calibrated Quinton Corval 400 (Quinton Instruments Inc., San Jose, CA, USA) electronically braked cycle ergometer using a linear mode. The TTE protocol was the same as in the study by Bergstrom et al. [14]. It may be possible that a lower cadence (70 rpm) may lead to differences in these studies. The results obtained by Wright et al. [28] demonstrated that the 3MT_lin_ may be reliable, but does not provide a valid estimation of CP. The values of CP were higher and W’ was lower compared to TTE performed on separate days and contained three tests to exhaustion at 80%, 100%, and 105% MAP at a preferred cadence and was calculated from the 1/time and linear-TW models. 

This test still requires two visits to the laboratory. Constantini et al. [17] examined the possibility of determining the CP and W’ using a linear mode within a single test, containing an incremental test followed by a 3MT_lin_ trial with a 20-min recovery period (15 passive + 5 active). This was performed on an electronically braked cycle ergometer (Excalibur Sport, Lode BV, Groningen, the Netherlands). The results showed that the CP and W’ were not significantly affected by the preliminary incremental test and provided an accurate and valid method. 

#### 4.2.2. The Alternative Test (3MT_%BM_)

The next possibility is to use the 3MT test within a single visit with the resistance set by (% BM. This can be calculated by the following equation: in the case of 3.5% body weight = 0.035 × bodyweight in kilograms. Bergstrom et al. [13] set the resistance to 3.5% and 4.5% BM on a Monark cycle ergometer (model818). The CP values of the 4.5% and 3.5% tests were not significantly different as compared to TTE (protocol given above), but the 3.5% test underestimated AWC by 38%. The results demonstrated that the 4.5% test was more accurate than the 3.5% test compared to the original CP test and 3MT_lin_. Different approaches, but similar results, have been confirmed by Clark et al. [16], comparing the 3MT linear factor to 3MT using % BM. Resistance was divided by fitness level to: 3% BM = recreationally active, 4% BM = aerobic and anaerobic athletes, and 5% BM = endurance athletes. The 3MT_%BM_ followed an exhaustive, square-wave bout at 10% above CP to provide the correct determination of the CP. All tests were performed on an electronically braked cycle ergometer (Lode Excalibur, Groningen, The Netherlands). This study confirmed that the % BM protocol is similar to the linear factor and may lead to a simplification of testing.

These results are in line with those obtained by Dicks et al. [18]. The precision of the % BM protocol was strongly correlated with the linear factor (*r* = 98). This protocol required the self-reporting of physical activity rating (PA-R), and the 3MT_%BM_ protocol involved a series of computations prior to the test. Power evoking maximum oxygen uptake (W_peak_) and power evoking GET (W_GET_) were estimated based on a percentage of W_peak_ from normative data using the following criteria. Participants with a PA-R of <8 are 60% W_peak_, and participants with a PA-R of 8 are 65% W_peak_. The average of W_GET_ and W_peak_ is used to estimate 50% Δ, then the PO is used to calculate a % BM. All tests were carried out on an electronically braked cycle ergometer (Lode Excalibur, Groningen, The Netherlands). The 3MT_lin_ and 3MT_%BM_ were similar to the TTE calculated from 1/time or linear-TW. This is corroborated by previous studies and confirms that % BM provides a more practical method. 

#### 4.2.3. The Alternative Test (3MT_iso_)

The 3MT test may be used in another alternative way, i.e., an isokinetic mode with fixed cadence. Karsten et al. [22] compared TTE containing three trials at work rates of 80%, 100%, and 105% MAP with preferred cadence calculated using a linear-TW, 1/time model with 3MT_iso_ where the resistance on the pedal was provided by the SRM ergometer and cadence was maintained at the preferred level throughout the whole test. The results showed that the values of the isokinetic mode do not provide a valid measure. The CP was significantly higher in comparison with the traditional protocol. Contrary to what was reported in the previous study, the results of Wright et al. [28] indicated that the values of CP provided a reliable measure but W’ was significantly lower compared to the TTE protocol, which was performed on separate days, containing three tests at 80%, 100%, and 105% MAP with the preferred cadence, calculated using 1/time, linear-TW. This protocol was carried out using an electronically braked cycle ergometer (Excalibur Sport, Lode, Groningen, The Netherlands). During this test, subjects cycled at their preferred cadence and were unable to cycle faster than the selected cadence and an increase in torque resulted in an increase in resistance.

#### 4.2.4. 3MT in More Available Conditions

All of these tests were carried out in the laboratory, but not all cyclists have access to a stationary mechanical cycle ergometer. Due to this, Clark et al. [15] introduced another opportunity to perform 3MT using a CompuTrainer cycle ergometer, which enables one to use this test even at home and on one’s own bicycle. In this test, subjects pedal with all-out effort for the entire three minutes and can change gears freely, and the CompuTrainer auto-adjusts resistance according to the cadence of the subject. The estimation of CP and W’ from this method was compared with constant-load trials, where the PO were calculated using the power–inverse time model with an estimated limit of 180, 360, or 540 s. The actual time with three different PO was used to calculate the CP and W’ with 1/time and the linear-TW model. The results supported the validity of the 3MT on a CompuTrainer. This test on the CompuTrainer should be performed only by regular cyclists. The protocol requires the subject to shift gears to match the intensity they can tolerate, and amateur cyclists are not able to perform this task effectively. The problem with the self-selection of flywheel torque for the 3MT is that subjects may pace themselves. Pacing weakens the validity of determination of CP and W’, with the underestimation of W’ and overestimation of CP [15]. In general, the 3MT protocol is recommended for a trained population or individuals with a preferred cadence ≥80 rpm [18].

#### 4.2.5. Prediction Validity of CP Using 3MT_lin_

CP has a predictive significance but is rarely used to predict performance because its assessment is demanding and time-consuming, due to the necessity of several laboratory visits. The efficacy of the 3MT in predicting cycle time-trial (TT) performance was examined by Black et al. [8]. The CP was derived from 3MT_lin_ with fixed resistance where the PO was equivalent to 50% of the difference between GET and VO_2max_ at the subject’s preferred cadence on an electronically braked cycle ergometer (Lode Excalibur Sport, Groningen, The Netherlands) with a ramp incremental test. The 16.1 km road trial was conducted on participants’ own road racing bike without any aerodynamic equipment. The results confirmed that the CP determined from 3MT_lin_ was significantly correlated with 16.1 km TT trials. These results provided evidence for the ecological validity and relevance of 3MT_lin_. This means that the determination of CP may provide a useful addition to the battery of tests used to track fitness and predict athletic performance. 

A limitation of the present protocol is that a CP validation test is not usually included to ensure that a physiological steady state had been established [37]. However, this is a common limitation within the literature, and it should also be noted that the original research by Vanhatalo et al. [33] on the 3MT did not include a CP validation test [29].

### 4.3. Field Testing Protocols 

Many coaches and sports scientists prefer field tests over laboratory tests because of the high ecological validity. Field tests may provide a better approach and are more applicable to practitioners and their daily work because of their higher specificity to sports performance. This protocol can be easily integrated into the training routine and may save more time than laboratory tests. The traditional method of CP uses a constant work approach in the laboratory to measure time to exhaustion, but it seems to be less reliable compared to fixed-distance or fixed-duration tests. Tests over a fixed duration or distance reflect competitive time-trial performance and might increase the ecological validity [27]. However, field tests may be viewed as less reliable than laboratory tests due to the lower level of control over environmental variables [38]. The estimation of CP in cycling has a tradition based on laboratory tests, while other sports use the field-based determination of the related parameter critical velocity (equivalent to CP, measured in m/s) more often, such as running or swimming [19]. In swimming, for example, only two performances over 200 m and 400 m are required [39].

#### 4.3.1. Comparing Field and TTE Protocols 

Field testing may be performed over a fixed time with a duration ranging from 3 to 12 min. Karsten et al. [19] used a protocol of 3, 7, and 12 min in an outdoor velodrome. These tests were completed on separate days in random order. This test started from a standing position with the aim of riding around the velodrome as fast as possible in each test. Capillary blood samples were taken and post-test blood lactate of ≥8 mmol/L or heart rate (HR) within 10 beats of an age-predicted maximum of HR indicated the attainment of VO_2max_ for acceptance of a successful all-out test. Field testing was compared to a TTE trial, which included three tests to exhaustion on separate days at work rates of 80%, 100%, and 105% MAP at the preferred cadence of the subject. Each protocol was performed on a road bike (Raleigh Airlite, UK) equipped with a PowerTap Elite wheel (CycleOps, Madison, WI, USA) and a magnet for direct cadence measurement. In the laboratory, it was attached to a CompuTrainer (RacerMate, Seattle, DC, USA). Linear and 1/time mathematical models were used for both laboratory and field testing. The results of the field conditions provided a valid determination of CP with strong correlations (r = 0.97–0.98), but a significant difference was found in the estimation of W’. 

Similar results were obtained in the research by Karsten et al. [23], in which they also compared TTE and field protocols. The TTE tests were performed at work rates equivalent to 80%, 100%, and 105% MAP with a 30-min recovery period. For all testing, participants used their own road bike with a PowerTap Elite wheel (CycleOps, Madison, WI, USA) and a magnet for direct cadence measurement. For laboratory testing, the road bike was attached to a CompuTrainer (RacerMate, Seattle, DC, USA). Environmental conditions were not standardized for ecological validity. In the field, subjects were required to record their training and racing activities with PowerTap. This research contained three experimental trials lasting 12, 7, and 3 min. The first experiment included a recovery period of 30 min, cyclists performed trials fully rested in the second experiment, and experiment three used the highest three PO values of all training and racing files for the determination of CP and W’. The 1/time model was used for calculation of CP and W’. All experimental protocols had a good level of agreement between the laboratory and the field. This demonstrates the approach that CP can be determined in controlled (planned maximal effort) and uncontrolled (data from training and performance) conditions. The first experiment contained a high level of agreement and low prediction errors, the other two experiments (2, 3) resulted in a lower level of agreement and higher prediction errors, but despite this there is the advantage of more integrated usage in the training schedule of riders. The research by Leo et al. [31] also compared controlled and uncontrolled conditions of the test during a competitive season among cyclists. CP test contained 2-, 5-, and 12-min maximal efforts in a randomized order in the field condition on a standardized uphill climb with an average gradient of 5.5%. Participants should maintain the highest possible effort at a cadence between 80 and 100 rpm, using 1/time mathematical model for CP calculation. PO in the field (uncontrolled) was recorded using a standardized crank system (SRAM Red; Quarq/SRAM, Spearfish, SD). The same 2-, 5-, and 12-min maximal mean PO values during every period were identified using a cycling software platform. Pre-season CP tests (controlled) showed good agreement with in-season derived CP field (uncontrolled) but not W’ field (uncontrolled). 

These results are in line with those obtained by Triska et al. [27], in which the values of CP and W’ were correlated between field and laboratory testing (r = 0.613 and r = 0.718). Despite the fact that there are no significant differences, a high random variation does not support the use of tests interchangeably. This study used a different protocol. Field tests were carried out for 12, 6, and 2 min on a flat road separated by a passive recovery time of 30 min with a preferred gear ratio and cadence. This was performed on the subject’s own bike using a mobile power meter (Schoberer Rad Messtechnik, Jülich, Germany). The TTE tests were not performed at a constant work rate but in the cadence-dependent mode where the PO increases with an increase in cadence on the Lode Excalibur Sport ergometer. The resistance was set to a linear factor so that the participants would attain the pre-determined work rate of 70% ∆ (i.e., Δ 70% of the difference between the work rate at VT and VO_2max_), 98%, or 110% P_max_ at a cadence of 100 rpm and calculated as linear factor = power output/cadence^2^. The exhaustion should appear within 2–15 min at that work rate. The passive recovery time was 30 min between each trial. The mathematical models: 2-hyp, linear, and 1/time were used for both protocols. However, monitoring cycling performance should be sustained in the laboratory due to valid and reliable results compared to fixed time/distance tests under the field conditions. 

#### 4.3.2. Predictive Validity of Determining CP in the Field 

These studies evaluated the precision of determining CP under field conditions as compared to the laboratory. A study by Nimmerichter et al. [26] attempted to assess the predictive validity of CP and W’ on cycling performance under field conditions. The trials consisted of a duration of 1, 4, and 10 min and 2, 7, and 12 min performed in order from the shortest to the longest separated by 60 min of recovery. All trials were carried out on the subject’s own bike with an SRM professional power meter (Schoberer Rad Messtechnik, Jülich, Germany), which measured PO at a rate of 1 Hz. During all trials, subjects selected a cadence between 90 and 100 rpm. Hyp-3 and 1/time were chosen as mathematical models. The last test was a predictive time trial lasting 20 min (TT20). PO during the TT20 was predicted from the models using this equation:*P* = W’/1200 + CP.(8)

The results demonstrated that the best mathematical model appeared to be 1/time using 2-min, 7-min, and 12-min trials. Both parameters CP and W’ predicted TT20 power with a prediction error <2%.

Field testing for cyclists is still something new and has its limitations. The testing protocols contain three trials with a fixed duration ranging from 1 to 12 min with a different recovery time (30 min, 60 min, or full rested). The main data on the PO are collected by power meters. These tests may offer a more ecologically valid and less expensive method of CP estimation but should be used for more experienced cyclists. However, the study by Karsten et al. [19] presented data from a small sample and the study should be replicated with a larger group of participants. The other study by Karsten et al. [23] recommended analyzing training-related changes in CP throughout the racing season by applying the field-test protocols of 12, 7, and 3 min or collecting data to obtain three PO values of training. 

This future recommendation was fulfilled by Leo et al. [31].

The performance profile of professional cyclists, U23, was studied during the competition season. CP and W’ was obtained during training and race. This study confirms the future usage of CP in real practice among professional cyclists. It leads to the conclusion that field tests may have great potential and it would be appropriate to focus more on them. 

### 4.4. Practical Recommendation 

All tests included in this review (Table 3) are summarized according to specific criteria. We intend to assist practitioners with applying the tests in natural sports conditions. Therefore, we set time efficiency, professional competence, and technical requirements to address actual applicability. We rate each test on whether or not it meets chosen criteria. The criteria aim to reflect the demands and practicality of each test. We believe that this a simple overview that could help with the selection of the tests, according to the information in Figure 3, which is a more theoretical guideline for each test.

## 5. Conclusions

In summary, testing methods offer two main approaches—traditional and 3-min all-out protocols. Researchers are trying to save time, simplify the methods, and make them more accessible. The most widely used protocols are the TTE and 3MT_lin_. These tests may be considered the most reliable and valid. Other tests, such as TT or 3MT_iso_ or 3MT_%BM_, are still not applied that often. The last option is field testing, which is not yet completely standardized. We suggest that future research should further examine field testing and try to make it more reliable and valid. It would be further desirable to focus on a different testing approach than the bicycle ergometer. 

## Figures and Tables

**Figure 1 ijerph-19-07589-f001:**
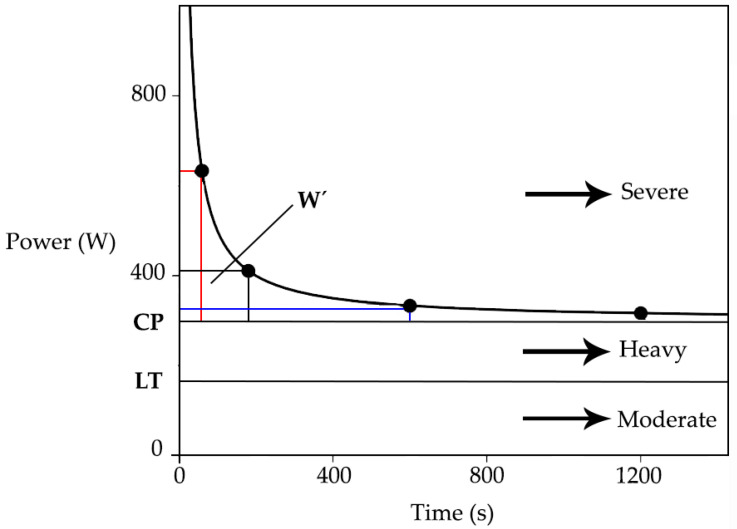
The power–time (P-t) relationship for high-intensity exercise.

**Figure 2 ijerph-19-07589-f002:**
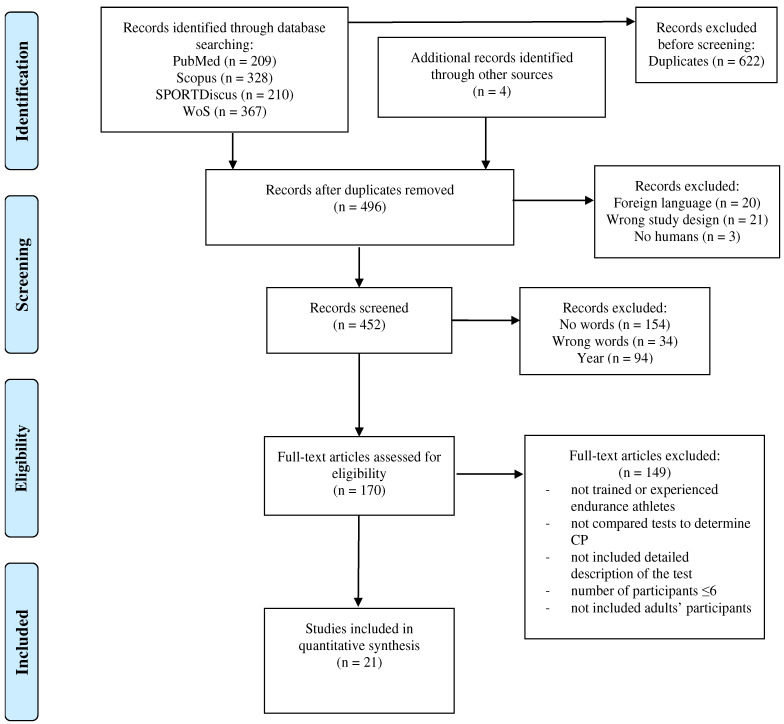
PRISMA flow diagram of the search strategy.

**Figure 3 ijerph-19-07589-f003:**
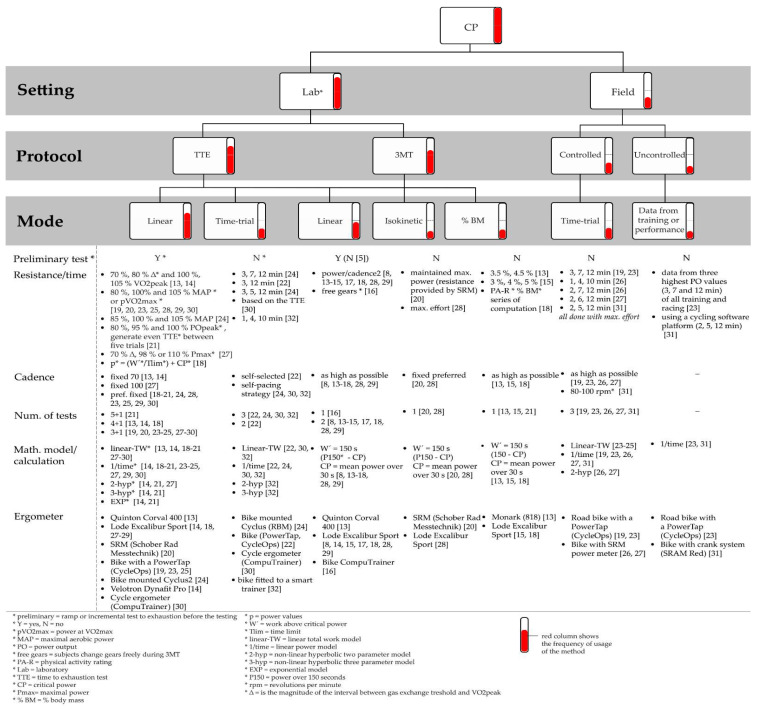
Scheme of all methods from studies included in this review [8,13,14,15,16,17,18,19,20,21,22,23,24,25,26,27,28,29,30,31,32]. (* explanation of the abbreviation).

**Table 1 ijerph-19-07589-t001:** Describing participants from all included studies.

Author	Year	*n*	Sex (F, M)	Age (Mean ± SD)	Subjects	VO_2max_ (mL·kg^−1^ min^−1^, * L·min^−1^)
Bergstrom et al. [13]	2012	12	6 F, 6 M	23.2 ± 3.5	Moderately trained	42.97 ± 7.42
Bergstrom et al. [14]	2014	9	5 F, 4 M	23 ± 3	Recreationally trained	* 3.01 ± 0.58
Black et al. [8]	2014	10	M	33.8 ± 8.2	Club-level cyclists	60 ± 4
Clark et al. [16]	2013	15	12 F, 3 M	22 ± 5	Varied fitness level	No
Clark et al. [15]	2016	10	3 F, 7 M	26 ± 9	Experienced cyclists	51.4 ± 6,05
Clark and Macdermid [32]	2021	10	M	25 ± 5	Elite cyclists	71.9 ± 5.9
Coakley and Passfield [30]	2018	17	M	31 ± 9	Trained cyclists	60.4 ± 8.4
Constantini et al. [17]	2014	12	6 F, 6 M	26 ± 4	Recreationally trained	51 ± 11.9
Dicks et al. [18]	2016	12	3 F, 9 M	27 ± 9	Varied fitness level	53.52 ± 8.02
Karsten et al. [20]	2014	13	1 F, 12 M	33 ± 7	Elite cyclists	* 5.18 ± 0.87
Karsten et al. [19]	2014	14	2 F, 12 M	40 ± 7	Trained cyclists	* 3.8 ± 0.5
Karsten et al. [23]	2015	11	M	32 ± 8	Recreationally competitive cyclists	51.4 ± 9.8
Karsten et al. [25]	2017	9	M	33 ± 8	Recreational cyclists	* 3.9 ± 0.4
Karsten et al. [24]	2018	12	M	39 ± 9	Moderately trained cyclists	54.7 ± 9.6
Leo et al. [31]	2021	13	M	21.1 ± 1.2	Professional cyclists	73.8 ± 1.9
Maturana et al. [21]	2018	13	4 F, 9 M	26 ± 3	Recreational or competitive cyclists	60.4 ± 5.9
Nimmerichter et al. [26]	2020	10	M	31.4 ± 5.8	Trained cyclists	No
Simpson and Kordi [22]	2017	8	1 F, 7 M	31 ± 4	Competitive amateur cyclists	No
Triska et al. [27]	2015	10	M	26.2 ± 4.1	Competitive cyclists	63.2 ± 5.5
Wright et al. [28]	2017	12	M	32 ± 6.6	Trained cyclists	* 4.4 ± 0.5
Wright et al. [29]	2019	10	M	30 ± 5	Trained cyclists	* 4.7 ± 0.4

F = Female, M = Male, SD = standard deviation, No = value not mentioned, * values measured in liter per minute.

**Table 2 ijerph-19-07589-t002:** Comparison of CP and W’ from different mathematical models using TTE and TT tests.

Authors		CP_EXP_	CP_3-hyp_	CP_2-hyp_	CP_linear-TW_	CP_1/time_
Bergstrom et al. [14]	CP	>3-hyp, 2-hyp, linear-TW, 1/time	<EXP, linear-TW, 1/time, 2-hyp	>3-hyp, <EXP, 1/time, linear-TW	>3-hyp, 2-hyp, <EXP, 1/time	>linear-TW, 2-hyp, 3-hyp, <EXP
	W’	—	>1/time, linear-TW, =2-hyp	>1/time, linear-TW, =3-hyp	<3-hyp, 2-hyp, 1/time	<3-hyp, 2-hyp, >linear-TW
Clark and Macdermind [32]	CP	—	<linear-TW, 1/time, 2-hyp	>3-hyp, <linear-TW, 1/time	>3-hyp, 2-hyp, <1/time	>linear-TW, 2-hyp, 3-hyp
	W’	—	>1/time, linear-TW, 2-hyp	>1/time, linear-TW, <hyp-3	>1/time, <3-hyp, 2-hyp	<3-hyp, 2-hyp, linear-TW
Maturana et al. [21]	CP	>3-hyp, 2-hyp, linear-TW, 1/time	<EXP, linear-TW, 1/time, 2-hyp	>3-hyp, <EXP, 1/time, linear-TW	>3-hyp, 2-hyp, <EXP, 1/time	>linear-TW, 2-hyp, 3-hyp, <EXP
	W’	—	>1/time, linear-TW, 2-hyp	>linear-TW, 1/time, <3-hyp	>1/time, <2-hyp, 3-hyp	<3-hyp, 2-hyp, linear-TW

CP = critical power, W’ = work above CP, Linear-TW = linear total work, 1/time = linear 1/time, 2-hyp = 2-parameter hyperbolic, 3-hyp = 3-parameter hyperbolic, EXP = exponential.

**Table 3 ijerph-19-07589-t003:** Summarization and evaluation of individual tests according to given criteria.

Setting	Protocol	Mode	Time Efficiency ^1^	Professional Competence ^2^	Technical Requirements ^3^
Laboratory	TTE	Linear	YES	YES	YES
		Time-trial	YES/NO *	YES	YES
	3MT	Linear	YES/NO *	YES	YES
		Isokinetic	NO	YES	YES
		% BM	NO	YES	YES
Field	Controlled	Time-trial	NO	YES	YES
	Uncontrolled	Data from training or performance	NO	NO	YES

^1^ Time efficiency means the time needed to complete one testing procedure (i.e., one laboratory visit or testing in the field on one occasion). If the number of testing procedures is greater than one and requires additional measurements before testing CP (e.g., preliminary test to obtain VO_2max_, measuring of body weight, anthropometry diagnostic using dual-energy X-ray absorptiometry), it is marked YES, otherwise NO. ^2^ Professional competence implies that an educated person (e.g., sports physician or cardiologist or sports scientist or fitness specialist) is necessary to complete the test. If such a person is needed, it is marked YES, otherwise NO. ^3^ Technical requirements reflect any demands for additional equipment (e.g., wattmeter, bicycle ergometer, or other devices that are not commonly available). If such equipment is required, it is marked YES, otherwise NO. * YES/NO: As indicated by [22,24], either consecutive days or a one-day testing procedure are possible.

## Data Availability

The data presented in this study are openly available in: [WoS, Scopus].

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
