# Peer review of "Determination of Critical Power Using Different Possible Approaches among Endurance Athletes: A Review"

_ijerph, 2022, doi:10.3390/ijerph19137589_

Round 1

Reviewer 1 Report

Critical power is an important parameter of aerobic function and is the highest average effort  without fatigue. This review found that the most widely used tests were the 3-minute all-out test set in a linear mode and the traditional protocol time to exhaustion and so on.  These results are of great theoretical significance and practical value for determination and work of critical power. Minor revision can be published in International Journal of Environmental Research and Public Health. However, there are some major issues need to be improved: However, there are some major issues need to be improved:

1.Introduction: Recent research progress is lacking

2. Materials and Methods:Short literature review years and less content;

3. References: References from 2002 to 2020, but from 2012 to 2021 in review;As a review paper, the most recent research progress is critical, please supplement the 2021 and 2022 references.

Reviewer 2 Report

Unfortunately, while the concept and intention of the review is interesting and relevant to the field of sports science, my opinion is that the review is not systematic enough to truly appreciate the quantity of sports science literature concerning this topic that has developed over the last few decades.

The authors do not provide sufficient rationale for why they have limited the search to the last 10 years, to 'categories' within only two databases, and articles without keywords or keywords that are similar but not exact. This I feel undermines the integrity of the review and potential to be definitive about the current approaches to testing critical power.

In addition to this main concept, the manuscript has frequent grammatical errors and issues with sentence and paragraph construction that make it difficult to follow as a reader. I also feel that the introduction would benefit from a conceptual figure of the critical power concept - as it is a hard concept to grasp with only literal description.

While the review has merit, I feel that fundamentally it would require a more extensive search of a greater number of databases, with a greater year span and less restrictive exclusion criteria to truly capture the evolution of the field and measures use to capture critical power. 

Thank you. 

Reviewer 3 Report

Dear Author, 

Please, find my comments attached.

Round 2

Reviewer 2 Report

1. I applaud the authors on their addition of two databases (PubMed, SPORTdiscus) to the original two databases (Scopus, Wos) and updating the search criteria, as well as their clarification of exclusion criteria.

2. While this strengthens the review rigour, the authors have not provided an adequate justification for why the search was limited to the last 10 years. Providing a reference to a book which actually shows a 13-year search does not provide ample justification for limiting the search period to 10 years, nor does the rationale of looking at recent knowledge, if there is no valid reason why knowledge prior to this point would be invalid. A systematic review should search all relevant literature for the concept, unless there is a clear rationale for limiting the search to a given year such as a previous review (for which best practice is to include the previous findings in the updated review) or the the creation of the concept. ('critical power' appears to have been defined in a seminal paper by Hill et al 1993). The authors should extend their search back to a relevant time period (such as the birth of the concept) with stronger justification for the timespan and/or acknowledgement of previous reviews in the literature (if existent). 

3. The grammar throughout the manuscript, mainly in the introduction, discussion and conclusion sections requires moderate revision. The authors needs to define and spell out all units and/or acronyms at first use (i.e. kj, VO2max, etc) regardless of their common usage. I applaud the authors for fixing many of the acronyms (i.e. Power output and PO) but there are still many undefined. The general structure of the sentences throughout these sections still have many small grammatical errors and improper attribution evident.

I.e. "Endurance sports are exposed to a fast pace, but even a 29 slight increase in pace may lead to a reduction in the tolerable duration of exercise. "

 - ' Endurance sports' are not exposed to a fast pace, more so the athletes. 

I.e. 'CP can help with racing strategy. An example is provided by two runners that have 59 an identical gas exchange threshold (GET), which is used as an index of anaerobic 60 threshold [9] and VO2max values. '

 - Likewise, Understanding 'CP' can help with racing strategy. An example is not 'provided by', it may be more appropriate to say 'This concept is best explained through considering the performance of two runners with an identical gas threshold....'

There are numerous examples of this throughout the manuscript which require revision. The use of a native English speaker to revise has improved the manuscript somewhat, but moderate revision is still required throughout for clarity and flow. 

4. The discussion section is extensive and in large sections simply restates the protocols of each study, with sub-optimal integration of these findings into a coherent overview. I believe that the type of review conducted by the authors is at times unclear - is it a systematic review as indicated by the use of the PRISMA process for searching and flow chart, or is it a scoping review indicated by the broad overview of tests provided, with limited of analysis of the internal validity of tests, or bias of reporting in each study. The authors try to make assertions about the validity of these tests in their conclusion, yet provide an overview in the results/discussion without presenting measures of internal validity or discussing the risk of bias of each included study.

5. The introduction would still benefit from a conceptual figure of critical power (and related concepts such as W'), given the technical nature of its explanation and importance to understanding the paper. Additionally, a clear table summarising advantages and disadvantages of each tool in the results/discussion would help to clarify the exceptionally long and dense results section. 

Thank you. 

Reviewer 3 Report

Dear Authors,

In my opinion, you made satisfactorily all the substantive corrections in the manuscript which I asked for.

My recommendation is to accept this manuscript in present form.

Sincerely yours,

Reviewer 3

Author Response

Dear Reviewer 3, 

Thank you again for all the comments. We greatly appreciate all the work and advice that has been provided.

Best regards

Authors